# Ionic Liquid Catalyzed Per-*O*-Acetylation and Benzylidene Ring-Opening Reaction

**Mayur Thul** [1], **Yao-Peng Wu** [1], **Yi-Jyun Lin** [1], **Shu-Lin Du** [2], **Hsin-Ru Wu** [3], **Wen-Yueh Ho** [4,*] **and Shun-Yuan Luo** [1,*]

1   Department of Chemistry, National Chung Hsing University, Taichung 402, Taiwan; mayurthul4@gmail.com (M.T.); grandy.hchs@gmail.com (Y.-P.W.); jiun16899@yahoo.com.tw (Y.-J.L.)
2   Department of Chemistry, Shu Guang Girl's Senior High School, Hsinchu 300, Taiwan; miradu0413@gmail.com
3   Instrumentation Center, MOST, National Tsing Hua University, Hsinchu 30013, Taiwan; hrwu@mx.nthu.edu.tw
4   Department of Cosmetic Science and Institute of Cosmetic Science, Chia Nan University of Pharmacy and Science, Tainan 717, Taiwan
*   Correspondence: rickho@mail.cnu.edu.tw (W.-Y.H.); syluo@dragon.nchu.edu.tw (S.-Y.L.); Tel.: +886-4-22840411 (S.-Y.L.)

**Abstract:** Tunable aryl imidazolium ionic liquids acting as Brønsted acid ionic liquids were found to be efficient catalysts for per-*O*-acetylation and reductive ring opening of benzylidene acetals. This method requires a truly catalytic amount of the least expensive available ionic liquids that are water-stable and reusable and also stable at room temperature. The reactions were obtained in one hour with good to excellent yields. These reactions can form C−O and C−H bonds with a high atom economy. Furthermore, the ionic liquid is an anomeric selective catalyst in per-*O*-acetylation and reductive ring opening of benzylidene acetals of sugar moieties.

**Keywords:** per-*O*-acetylation; ring-opening; ionic liquid; reuse; Brønsted-Lowry acid

## 1. Introduction

Regioselectively functionalized carbohydrates have recently become an important development in glycochemistry and glycobiology. Sugar molecules consist of several hydroxyl groups that are difficult to employ selectively as they have similar reactivity profiles. Therefore, methodologies that accomplish potent protection and selective alteration of monosaccharides are fundamental synthetic appliances in organic chemistry [1]. The use of distinctive hydroxyl groups in selective protection of carbohydrate molecules is a key step in the chemical synthesis of complex carbohydrates. Per-*O*-acetylation of sugars is an essential intermediate in carbohydrate transformation and synthesis [2]. In this respect, the ring-opening of cyclic benzylidene acetals comparable to *O*-benzyl ethers, in a regioselective aspect, is a favorable path owing to the ease of formation of the acetal, as well as the well-established quality of the benzyl ether protection [3]. One major and important transformation in carbohydrate chemistry is the acetylation of monosaccharides. In particular, per-acetylated carbohydrates are imperative and convenient intermediates in the chemical synthesis of complex carbohydrates, particularly for chemical glycosylation. Per-*O*-acetylated sugars can be employed directly as glycosylation donors [4,5]. Per-*O*-acetylation is one of the most commonly utilized reactions in carbohydrates, mainly for primary protection of sugars. The acetylation reaction is commonly performed using acetic anhydride as the reagent and an array of catalysts. However, the surplus of acetic anhydride as a solvent causes tedious work in the neutralization process. A ring conformation that can fix benzylidene acetals

either oxidatively or reductively is a favorable protective group in carbohydrate [6,7]. The reductive ring opening of benzylidene acetals is an essential reaction in this context and has been very useful in the selective manipulation of neighboring hydroxyls in polyols. The first reagent utilized for affecting this conversion was LiAlH$_4$–AlCl$_3$ [8–14]. The benzylidene sugar can form two regiospecific benzyl ethers by the reductive ring-opening reactions [15–18]. Various Lewis acid catalysts, such as InCl$_3$ [19], In(OTf)$_3$ [20], ZnCl$_2$ [21], Sc(OTf)$_3$ [22], TMSOTf [23], Cu(OTf)$_2$ [24], and triphenyl carbenium tetrafluoroborate [25], have been introduced for the acid-catalyzed per-*O*-acetylation and reductive ring opening of benzylidene acetals of sugars. However, these catalysts also undergo significant shortcomings, such as an expensive reagent, long reaction time, and harsh reaction conditions. Solid acid catalysts can relieve these problems, as they are cheap and allow the straightforward deportation of the catalysts from the reaction system [26–30]. The function of ionic liquids (Ils) as environmentally favorable reaction solvents for synthesis and catalysis has received broad recent consideration [31–34]. Per-*O*-acetylation of sugars catalyzed by ionic liquids such as dicyanamide based ionic liquid, zinc-based ionic liquid, and dialkyl imidazolium benzoates ionic liquid were studied [35,36]. This work reports a systematic screen of selected ionic liquids (**Ia–If**, Scheme 1) The Brønsted–Lowry acid as catalysts for a per-*O*-acetylation of ᴅ-glucose and regioselective ring opening of benzylidene acetals at the *O*-4 position of glucopyranoside. These ionic liquid catalysts have, to the best of our knowledge, not been previously utilized in this transformation. The imidazolium-based ionic liquids are synthesized as a novel type of ionic liquid, which acts as a Brønsted acid catalyst for both per-*O*-acetylation and benzylidene ring-opening reactions, is moisture-insensitive, stable at room temperature and reusable.

**Scheme 1.** Tunable aryl imidazolium ionic liquid **Ia–If**.

## 2. Results and Discussion

Selective per-*O*-acetylation of ᴅ-glucose (**1a**) was explored as a model compound. Treatment of compound **1a** with acetic anhydride (10 equiv.) in the presence of ionic liquid **Ia** (0.1 equiv.) at room temperature proceeded to completion of the reaction in one hour (Table 1, entry 1). Under these conditions, the product 1,2,3,4,6-penta-*O*-acetyl-ᴅ-glucopyranoside (**2a**) was obtained in 99% yield respectively. Subsequently, ionic liquids **Ib**, **Id**, **Ie**, and **If** (Scheme 1) were investigated in this transformation. Under the same conditions as described above, these catalysts provided per-*O*-acetylation **2a** in good to excellent yields (Table 1, entry 2, 4, 5, and 6). However, **Ic** did not proceed with the reaction smoothly, and achieved a 29% yield of **2a** (Table 1, entry 3), because of the weak acidity of the **Ic.**

### 2.1. Per-O-Acetylation

The study of the ionic liquid suggested that **Ia** was the best catalyst for the per-*O*-acetylation reaction, providing **2a** in excellent yield. The reaction was routinely monitored using TLC and produced **2a** after applying purification compounds. Based on these preliminary results, the acetic anhydride loading was explored. To determine the minimum amount of acetyl reagent required, we used 7.5 equivalents and 6.0 equivalents of acetic anhydride with ionic liquid for 24 h (Table 1, entry 10). Increasing the temperature to 80 °C resulted in the formation of selective **2a** in an isolated yield of 92% in very short reaction time (Table 1, entry 11).

**Table 1.** Optimized condition of per-*O*-acetylation without solvent.

| Entry | IL | T (°C) | t (h) | Ac$_2$O (eq) | P (Yield) [a] |
|-------|----|--------|-------|--------------|---------------|
| 1 | **Ia** | 25 | 1 h | 10 | **2a** (99%) |
| 2 | **Ib** | 25 | 1 h | 10 | **2a** (98%) |
| 3 | **Ic** | 25 | 1 h | 10 | **2a** (29%) |
| 4 | **Id** | 25 | 1 h | 10 | **2a** (70%) |
| 5 | **Ie** | 25 | 1 h | 10 | **2a** (96%) |
| 6 | **If** | 25 | 1 h | 10 | **2a** (86%) |
| 7 | **Ia** | 25 | 1 h | 7.5 | **2a** (99%) |
| 8 | **Ia** | 25 | 1 h | 6.0 | **2a** (99%) |
| 9 | **Ia** | 25 | 1 h | 5.25 | **2a** (95%) |
| 10 | **Ia** | 25 | 24 h | 6.0 | **2a** (94%) |
| 11 | **Ia** | 80 | 1 h | 6.0 | **2a** (92%) |

[a] The yields are isolated yields.

The scope, limitations, and generality of the method were explored. Table 1 shows the per-*O*-acetylation of other important ᴅ-hexoses **1b**–**1d**, ᴅ-pentose **1e**, and disaccharide **1f** under this set of optimized conditions. ᴅ-galactose **1b** readily provided the corresponding penta-acetate **2b** quantitatively (Table 2, entry 1), whereas ᴅ-mannose **1c** afforded product **2c** in excellent yield 93% (Table 2, entry 2). Reducing the equivalent of acetic anhydride to 4.8 equivalents led to a similar result in the case of methyl glucopyranoside **1d**, and the expected compound **2d** was isolated in 99% yield (Table 2, entry 3). Equally, ᴅ-xylose **1e** led to exclusive formation of the corresponding product **2e** in 97% yield (Table 2, entry 4) and increasing the equivalent of acetic anhydride, the substrate **2f** produced compound **1f** in 96% yield (Table 2, entry 5), respectively.

**Table 2.** Ionic liquid (**Ia**) catalyzed the solvent-free per-*O*-acetylation of hexoses.

| Entry | SM | t (h) | Ac$_2$O (eq) | P (Yield) |
|-------|----|-------|--------------|-----------|
| 1 | **1b** | 1 h | 6 | **2b** (98%) |
| 2 | **1c** | 3 h | 6 | **2c** (93%) |
| 3 | **1d** | 1 h | 4.8 | **2d** (99%) |
| 4 | **1e** | 1 h | 4.8 | **2e** (97%) |
| 5 | **1f** | 1 h | 9.6 | **2f** (96%) |

*2.2. Reusability of TAIIL for Per-O-Acetylation of ᴅ-Glucose*

The anticipation of ionic liquid as a solvent replacement, especially on broad proportion application, counts on the usage of potent recycling to help blunt costs and curtail their environmental impact.

In the reaction of **1a** with acetic anhydride in the presence of **Ia** to form **2a**. After the filtration of the reaction mixture and the extraction of the product with ethyl acetate for ionic liquid, the ILs were concentrated, dried under a high vacuum, and reused. Figure 1 shows five cycles of the acetylation in ILs that are recovered and reused for further per-*O*-acetylation reactions. No significant loss of activity of these products was observed (91–99%). After reuse of ionic liquid **Ia**, $^{19}$F NMR confirmed that ionic liquid remains in the aqueous phase.

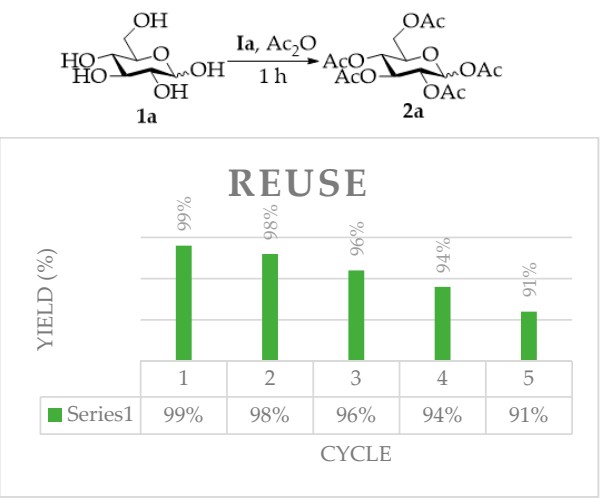

**Figure 1.** Recycling of **Ia** in the synthesis of penta-acetate D-glucose **2a**.

The plausible mechanism (Scheme 2) of the reaction appears to be through the acylium intermediate formed by the reaction of Ionic liquid and Ac$_2$O. The acylium intermediate would be able to acetylate the free alcohol moieties very efficiently. After starting material **1** reacted with the acylium intermediate to form **1'** the **Ia'** abstracts the proton from **1'** to form acetylated compound **2**.

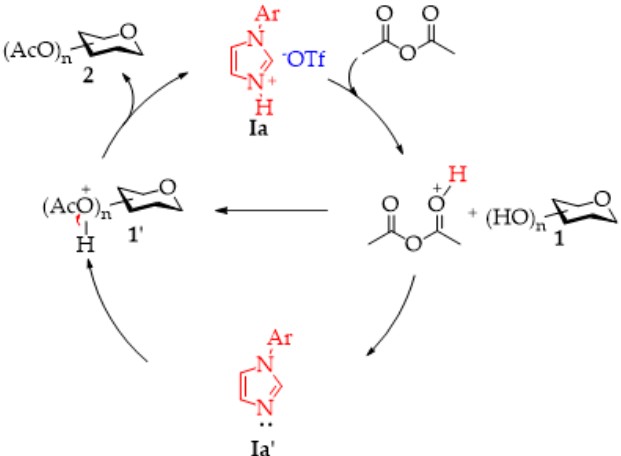

**Scheme 2.** A plausible mechanism of the per-*O*-acetylation reaction.

### 2.3. Reductive Ring Opening of Benzylidene Acetals

This investigation reports a systematic screen of selected ionic liquids (**Ia–If**, Scheme 1) as catalysts for a regioselective ring opening of benzylidene acetals at the *O*-4 position of hexopyranosides. Those ionic liquid catalysts have, to our knowledge, not previously been utilized in this transformation. The reaction conditions using the best ionic liquid catalysts were optimized to provide a robust and reliable procedure. The *O*-4 selective ring-opening reaction was selected for optimization. Initially, sugar **3a** was accepted as an excellent substrate. Firstly, **3a** with ionic liquid, **Ia** (1.0 equiv.), and triethylsilane (10.0 equiv.) were used in dichloromethane (DCM) at room temperature. The expected

product **4a** was assembled, then the reaction mixture was treated with tetra-*n*-butyl ammonium fluoride (TBAF, 11.0 equiv.) and acetic acid (AcOH, 11.0 equiv.) at room temperature for one hour, providing **4a** in 78% yield (Table 3, entry 1). Screening results of ionic liquids (Table 3, entry 2−6) indicated that **Ia** was superior to **Ib–If**, probably due to weak acidity of ionic liquid **Ib–If**.

**Table 3.** Optimized conditions of regioselective *O*-4 ring-opening reactions.

| Entry | Solvent | IL (eq) | t (h) | P (Yield) [a] |
|-------|---------|---------|-------|---------------|
| 1 | DCM | **Ia** (1.0) | 6 h | **4a** (78%) |
| 2 | DCM | **Ib** (1.0) | 6 h | **4a** (4%) |
| 3 | DCM | **Ic** (1.0) | 6 h | **4a** (6%) |
| 4 | DCM | **Id** (1.0) | 6 h | **4a** (8%) |
| 5 | DCM | **Ie** (1.0) | 6 h | **4a** (58%) |
| 6 | DCM | **If** (1.0) | 6 h | **4a** (4%) |
| 7 | ACN | **Ia** (1.0) | 1 h | **4a** (88%) |
| 8 | ACN | **Ia** (0.5) | 1 h | **4a** (90%) |
| 9 | ACN | **Ia** (0.25) | 2 h | **4a** (77%) |

[a] The yields are isolated yields.

Performing the reaction with ACN as the solvent yielded 88% of the product **4a** (Table 3, entry 7). Decreasing the ionic liquid catalyst to 0.5 and 0.25 equivalents yielded product **4a** in 90% and 77% yield respectively (Table 3, entry 8–9).

The optimized conditions were examined. Thus, compounds commonly used in carbohydrate chemistry with different protection groups were tested in this transformation. As indicated in (Table 4, entry 1), the corresponding product **4b** was obtained in excellent yield after increasing reaction time. Under the same optimized condition, **3c** was transformed into **4c** as the only product (Table 4, entry 2). In the subsequent reaction, with a longer reaction time of **8h**, **3d** was converted into **4d** with a 72% yield (Table 4, entry 3). However, the reaction of **3e** afforded **4e** as the final compound in good yield (Table 4, entry 4).

**Table 4.** Reductive ring-opening of various 4,6-*O*-benzylidene-ᴅ-hexopyranosides at room temperature in the presence of **Ia** as a catalyst.

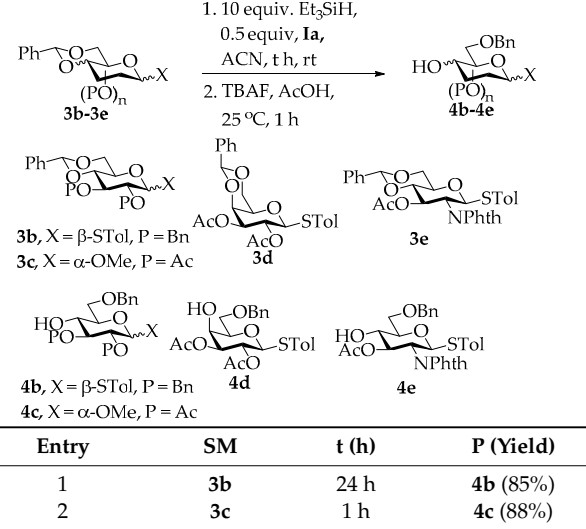

| Entry | SM | t (h) | P (Yield) |
|-------|-----|-------|-----------|
| 1 | **3b** | 24 h | **4b** (85%) |
| 2 | **3c** | 1 h | **4c** (88%) |
| 3 | **3d** | 8 h | **4d** (72%) |
| 4 | **3e** | 1 h | **4e** (82%) |

The plausible mechanism (Scheme 3) shows that acetals oxygen (*O*-4) **3** abstract a proton from ionic liquid **Ia** to form the intermediate **3′**, which should then have converted to intermediate **a**. Then, the addition of Et₃SiH to form the **3a**. Finally, the treatment of *tetra-n*-butylammonium fluoride (TBAF) and acetic acid with **a′** to obtain the desired product **4**, with the removal of triethylsilyl fluoride (TESF).

**Scheme 3.** A plausible mechanism of the benzylidene ring-opening reaction.

## 3. Materials and Methods

### 3.1. Synthesis of Ionic Liquid

The preparation of ionic liquids **Ia–If** (Scheme 4) is described below. Ionic liquids **Ia–If** were synthesized in two steps. The first step is the Ullmann-type coupling reaction: a combination of imidazole **A** and 1-iodo-4-nitrobenzene **B** in the presence of 10 mol % copper(II) acetate and cesium carbonate gave aryl imidazole **C₁**, **C₂**, **C₃**, **C₄** in 74%, 80%, 76%, and 88% yields respectively. In the second step, the aryl imidazole **C₁**, **C₂**, **C₃**, **C₄** was dissolved in ethanol and treated with triflic acid, methane sulfonic acid, or trifluoroacetic acid in an ice bath to yield acidic ionic liquids **Ia–If**. For the characterization of ionic liquids, we check the Hammett acidity function using crystal violet dye as a reacting dye to check the acidity of ionic liquid **Ia–If** (See Supplementary Materials).

**Scheme 4.** Synthesis of ionic liquids.

## 3.2. General Procedure

To a solution of **C$_1$**, **C$_2$**, **C$_3$**, **C$_4$** (1 equiv.) and triflic acid, trifluoro methanesulfonic acid, and trifluoroacetic acid (2 equiv.) in ethanol (0.5 mL/mmol). The reaction mixture was stirred for two hours in an ice bath. The solvent was dried and the reaction mixture was washed with ether several times. The solvent was dried to give **Ia–Ic** as a yellow solid and **Id–If** as a white solid.

*1-(4-Nitrophenyl)-1H-imidazole-3-ium trifluoromethanesulfonate* (**Ia**). $^1$H NMR (400 MHz, CD$_3$OD) δ 9.30 (m, 1H), 8.18 (m, 2H), 7.91 (dd, *J* = 10.0, 8.4, Hz, 1H), 7.76 (m, 2H), 7.54 (d, *J* = 3.6 Hz, 1H); HRMS (ESI, M + Na$^+$) calcd for C$_{16}$H$_{22}$O$_{11}$Na 190.0618, found 190.0616.

*1-(4-Nitrophenyl)-1H-imidazole-3-ium methanesulfonate* (**Ib**). $^1$H NMR (400 MHz, Methanol-d$_4$) δ 9.53 (m, 1H), 8.38 (m, 2H), 8.11 (m, 1H), 7.93 (m, 2H), 7.72 (m, 1H).

*1-(4-Nitrophenyl)-1H-imidazole-3-ium trifluoromethaneacetate* (**Ic**). $^1$H NMR (400 MHz, CD$_3$OD) δ 9.35 (s, 1H), 8.16 (m, 2H), 7.94 (s, 1H), 7.74 (m, 2H), 7.54 (s, 1H).

*Phenyl-1H-imidazole-3-ium trifluoromethanesulfonate* (**Id**). $^1$H NMR (400 MHz, CD$_3$OD) δ 9.42 (s, 1H), 8.06 (s, 1H), 7.75 (m, 3H), 7.64 (m, 3H); HRMS (ESI, M + Na$^+$) calcd for C$_{16}$H$_{22}$O$_{11}$Na 145.07653, found 145.07657.

*1-(p-Tolyl)-1H-imidazole-3-ium trifluoromethanesulfonate* (**Ie**). $^1$H NMR (400 MHz, CD$_3$OD) δ 9.37 (d, *J* = 1.2 Hz, 1H), 8.01 (d, *J* = 1.2 Hz, 1H), 7.74 (s, 1H), 7.59 (d, *J* = 7.6 Hz, 2H), 7.44 (d, *J* = 8.0 Hz, 2H), 2.44 (s, 3H); HRMS (ESI, M + Na$^+$) calcd for C$_{16}$H$_{22}$O$_{11}$Na 159.09247, found 159.09222.

*1-(4-Methoxyphenyl)-1H-imidazole-3-ium trifluoromethanesulfonate* (**If**). $^1$H NMR (400 MHz, CD$_3$OD) δ 9.30 (s, 1H), 7.96 (m, 1H), 7.73 (d, *J* = 1.6 Hz, 1H), 7.64 (dd, *J* = 6.8, 2.0 Hz, 2H), 7.14 (m, 2H), 3.87 (s, 3H); HRMS (ESI, M + Na$^+$) calcd for C$_{16}$H$_{22}$O$_{11}$Na 175.08719, found 175.08714.

## 3.3. General Information

The reactions were conducted in flame-dried glassware, beneath the N$_2$ atmosphere. Acetonitrile and dichloromethane were refined and dried from a secure purification system containing activated Al$_2$O$_3$. All reagents were obtained from good-value sources and were nor purified unless otherwise mentioned. Flash column chromatography was enforced to Silica Gel 60. Thin-layer chromatography (TLC) was performed on precoated glass plates of Silica Gel 60 F254 disclosure was accomplished by spraying with a solution of Ce(NH$_4$)$_2$(NO$_3$)$_6$ (0.5 g), (NH$_4$)$_6$Mo$_7$O$_{24}$ (24.0 g) and H$_2$SO$_4$ (28.0 mL) in water (500.0 mL) and heating on a hot plate. Optical rotations were measured at 589 nm (Na), $^1$H and $^{13}$C NMR were recorded with 400 MHz instruments. Chemical shifts are in ppm from Me$_4$Si generated from the CDCl$_3$ lock signal at δ 7.26. Infrared spectra were taken with a Fourier transform infrared (FT-IR) spectrometer using NaCl plates. Mass spectra were analyzed on an instrument with an ESI source.

*1,2,3,4,6-Penta-O-acetyl-D-glucopyranose* (**2a**). In a solution of **1a** (200 mg, 1.11 mmol), ionic liquid **Ia** (37 mg, 0.11 mmol) and anhydride (633 μL, 6.7 mmol) was stirred at 25 °C for one hour within the sealed tube. The mixture was quenched with water (30 mL) and extracted with ethyl acetate (3 × 30 mL). The combined organic layers were dried over anhydrous MgSO$_4$, filtered, and concentrated. It had been refined by chromatography on silica gel to give the required product **2a** (433 mg, 99%, α/β = 1/0.3) as a white solid. $R_f$ 0.53 (EtOAc/Hex = 1/1); mp 109–113 °C; [α]$^{29}_D$ +62.5 (*c* 1.0, DCM); IR (NaCl) ν 1752, 1647, 1371, 1221, 1147 cm$^{-1}$; $^1$H NMR (400 MHz, CDCl$_3$) δ 6.31 (d, *J* = 3.6 Hz, 1H), 5.69 (d, *J* = 8.4 Hz, 0.3H), 5.45 (t, *J* = 9.8 Hz, 1H), 5.23 (t, *J* = 9.8 Hz, 0.3H), 5.15–5.12 (m, 1H), 5.09 (d, *J* = 3.6 Hz, 1H), 5.06 (d, *J* = 3.6 Hz, 0.5H), 4.28-4.22 (m, 1H), 4.12-4.08 (m, 2H), 4.05(d, *J* = 3.6 Hz, 0.5H), 3.81 (ddd, *J* = 10.0, 4.4, 2.0 Hz, 0.3H), 2.16 (s, 3H), 2.09 (s, 1H), 2.07 (d, *J* = 2.8 Hz, 3H), 2.06 (s, 1H), 2.02 (s, 3H), 2.00 (s, 1H), 2.01 (d, *J* = 1.8 Hz, 3H), 1.99 (s, 3H); $^{13}$C NMR (100 MHz, CDCl$_3$) δ 170.6, 170.2, 169.6, 169.4, 168.7, 89.0, 69.8, 69.2, 67.9, 61.4, 20.9, 20.7, 20.6, 20.5, 20.4; HRMS (ESI, M + Na$^+$) calcd for C$_{16}$H$_{22}$O$_{11}$Na 413.1060, found 413.1050.

*1,2,3,4,6-Penta-*O*-acetyl-*D*-galactopyranose* (**2b**). In a solution of **1a** (200 mg, 1.11 mmol), ionic liquid **Ia** (37 mg, 0.11 mmol) and anhydride (633 μL, 6.7 mmol) was stirred at 25 °C for one hour within the sealed tube. The mixture was quenched with water (30 mL) and extracted with ethyl acetate (3 × 30 mL). The combined organic layers were dried over anhydrous $MgSO_4$, filtered, and concentrated. It had been refined by chromatography on silica gel to give the required product **2b** (425 mg, 98%) as a white solid. $R_f$ 0.50 (EtOAc/Hex = 1/2); mp 70–80 °C; $[\alpha]^{29}_D$ +80.8 (*c* 1.0, DCM); IR (NaCl) ν 1751, 1648, 1373 cm$^{-1}$; $^1$H NMR (400 MHz, CDCl$_3$) δ 6.35 (d, *J* = 1.6 Hz, 1H), 5.47 (d, *J* = 2.4 Hz, 1H), 5.31 (dd, *J* = 2.0, 1.2 Hz, 2H), 4.34–4.29 (m, 1H), 4.08 (d, *J* = 4.0 Hz, 1H), 4.06 (d, *J* = 4.0 Hz, 1H), 2.13 (s, 3H), 2.13 (s, 3H), 2.02 (s, 3H), 1.99 (s, 3H), 1.98 (s, 3H); $^{13}$C NMR (100 MHz, CDCl$_3$) δ 170.0, 169.9, 169.8, 169.6, 168.6, 89.4, 68.5, 67.2, 67.1, 66.2, 61.0, 20.6, 20.5, 20.4, 20.3; HRMS (ESI, M + Na$^+$) calcd for $C_{16}H_{22}O_{11}Na$ 413.1060, found 413.1056.

*1,2,3,4,6-Penta-*O*-acetyl-*α*-*D*-mannopyranose* (**2c**). In a solution of **1c** (200 mg, 1.11 mmol), ionic liquid **Ia** (37 mg, 0.11 mmol) and Ac$_2$O (633 μL, 6.7 mmol) was stirred at 25 °C for three hours within the sealed tube. The mixture was quenched with water (30 mL) and extracted with ethyl acetate (3 × 30 mL). The combined organic layers were dried over anhydrous $MgSO_4$, filtered, and concentrated. It was absolutely purified by chromatography on silica gel to give the desired product **2c** (401 mg, 93%) as colorless oil. $R_f$ 0.50 (EtOAc /Hex = 1/2); mp 70–80 °C; $[\alpha]^{29}_D$ +80.8 (*c* 1.0, DCM); IR (NaCl) ν 1751, 1648, 1373 cm$^{-1}$; $^1$H NMR (400 MHz, CDCl$_3$) δ 6.06 (d, *J* = 1.6 Hz, 1H), 5.32 (d, *J* = 2.4 Hz, 1H), 5.24 (t, *J* = 2.2 Hz, 1H), 4.29–4.22 (m, 1H), 4.14–4.00 (m, 3H), 2.16 (s, 3H), 2.15 (s, 3H), 2.07 (s, 3H), 2.03 (s, 3H), 1.98 (s, 3H); $^{13}$C NMR (100 MHz, CDCl$_3$) δ 170.3, 169.7, 169.4, 169.3, 167.8, 90.2, 70.2, 68.4, 68.0, 65.2, 61.8, 20.5, 20.4, 20.3, 20.29, 20.26; HRMS (ESI, M + Na$^+$) calcd for $C_{16}H_{22}O_{11}Na$ 413.1060, found 413.1056.

*Methyl-2,3,4,6-tetra-*O*-acetyl-*α*-*D*-glucopyranoside* (**2d**). In a solution of **1d** (200 mg, 1.11 mmol), ionic liquid **Ia** (34 mg, 0.1 mmol) and anhydride (463 μL, 4.9 mmol) was stirred at 25 °C for one hour within the sealed tube. The mixture was quenched with water (30 mL) and extracted with ethyl acetate (3 × 30 mL). The combined organic layers were dried over anhydrous $MgSO_4$, filtered, and concentrated. It was absolutely purified by chromatography on silica gel to give the desired product **2d** (370 mg, 99%) as a white solid. $R_f$ 0.60 (EtOAc/Hex = 1/2); mp 51–53 °C; $[\alpha]^{29}_D$ +46.7 (*c* 1.0, DCM); IR (NaCl) ν 2960, 1750, 1648, 1371, 1225 cm$^{-1}$; $^1$H NMR (400 MHz, CDCl$_3$) δ 5.45 (dd, *J* = 10.0, 9.6 Hz, 1H), 5.04 (dd, *J* = 10.2, 9.4 Hz, 1H), 4.93 (d, *J* = 3.6 Hz, 1H), 4.87 (dd, *J* = 10.2, 3.8 Hz, 1H), 4.24 (dd, *J* = 12.4, 4.4 Hz, 1H), 4.08 (dd, *J* = 12.4, 2.4 Hz, 1H), 3.96 (ddd, *J* = 10.0, 4.4, 2.2 Hz, 1H), 3.39 (s, 3H), 2.08 (s, 3H), 2.05 (s, 3H), 2.00 (s, 3H), 1.98 (s, 3H); $^{13}$C NMR (100 MHz, CDCl$_3$) δ 170.4, 169.9, 169.8, 169.4, 96.6, 70.6, 69.9, 68.3, 67.0, 61.7, 55.3, 20.5, 20.46, 20.4; HRMS (ESI, M + Na$^+$) calcd for $C_{15}H_{22}O_{10}Na$ 385.1111, found 385.1111.

*1,2,3,5-Tetra-*O*-acetate-*D*-xylopyranose* (**2e**). In a solution of **1e** (200 mg, 1.11 mmol), ionic liquid **Ia** (44 mg, 0.13 mmol) and anhydride (603 μL, 6.38 mmol) was stirred at 25 °C for one hour within the sealed tube. The mixture was quenched with water (30 mL) and extracted with ethyl acetate (3 × 30 mL). The combined organic layers were dried over anhydrous $MgSO_4$, filtered, and concentrated. It was absolutely purified by chromatography on silica gel to give the desired product **2e** (419 mg, 97%, α/β = 5/1) as colorless oil. $R_f$ 0.55 (EtOAc/Hex = 1/1); $[\alpha]^{29}_D$ +51.8 (*c* 1.0, DCM); IR (NaCl) ν 2924, 1743, 1649, 1370, 1211 cm$^{-1}$; $^1$H NMR (400 MHz, CDCl$_3$) δ 6.25 (d, *J* = 3.6 Hz, 1H), 5.70 (d, *J* = 6.8 Hz, 0.18H), 5.46 (t, *J* = 9.9 Hz, 1H), 5.36 (t, *J* = 5.0 Hz, 0.32H), 5.30–5.23 (m, 0.28H), 5.19 (t, *J* = 8.4 Hz, 0.26H), 5.06–5.00 (m, 2H), 4.35 (dd, *J* = 12.0, 4.0 Hz, 0.21H), 4.24 (d, *J* = 6.0 Hz, 0.14H), 4.14 (dd, *J* = 12.4, 4.8 Hz, 0.23H), 3.99 (dd, *J* = 12.0, 6.0 Hz, 0.22H), 3.93 (dd, *J* = 11.2, 5.9 Hz, 1H), 3.70 (t, *J* = 11.0 Hz, 1H), 3.52 (dd, *J* = 12.0, 8.4 Hz, 0.2H). 2.17 (s, 3H), 2.13-2.05 (m, 9H), 2.05 (s, 3H), 2.04 (s, 3H), 2.02 (s, 3H); $^{13}$C NMR (100 MHz, CDCl$_3$) δ 170.0, 169.7, 169.6, 168.9, 89.1, 69.2, 69.16, 68.5, 60.5, 20.8, 20.6, 20.6, 20.4; HRMS (ESI, M + Na$^+$) calcd for $C_{13}H_{18}O_9Na$ 341.0849, found 341.0842.

*1,2,3,6-Tetra-*O*-acetate-4-*O*-(2,3,4,6-tetra-*O*-acetyl-*β*-*D*-galactopyra-nosyl)-*D*-glu-copyranose* (**2f**). In a solution of **1f** (500 mg, 1.47 mmol), ionic liquid **Ia** (51 mg, 0.15 mmol) and anhydride (1.33 mL,

14. mmol) was added in the dealed tube and stirred at 25 °C for one hour. The mixture was quenched with water (30 mL) and extracted with ester (3 × 30 mL). The combined organic layers were dried over anhydrous MgSO$_4$, filtered, and concentrated. It was absolutely purified by chromatography on silica gel to give the desired product **2f** (955 mg, 96%, α/β = 1/0.22) as white solid. $R_f$ 0.44 (EtOAc/Hex = 3/2); mp 85–87 °C; $[\alpha]^{22}_D$ +55.8 (*c* 1.0, DCM); IR (NaCl) ν 2981, 2943, 1753, 1649, 1434, 1371 cm$^{-1}$; $^1$H NMR (400 MHz, CDCl$_3$) δ 6.25 (d, *J* = 3.6 Hz, 1H), 5.66 (d, *J* = 8.3 Hz, 0.2H), 5.46 (t, *J* = 9.8 Hz 1H), 5.35 (d, *J* = 3.6 Hz, 1.2H), 5.24 (t, *J* = 9.0 Hz, 0.2H), 5.12 (dd, *J* = 10.4, 8.0 Hz, 1H), 5.08–5.04 (m, 0.2H), 5.00 (dd, *J* = 10.4, 3.6 Hz, 1H), 4.95 (dd, *J* = 10.4, 3.2 Hz, 1H), 4.49–4.42 (m, 3H), 4.18–4.05 (m, 4H), 4.00 (ddd, *J* = 10.2, 3.8, 2.0 Hz, 1H), 3.88 (t, *J* = 6.8 Hz, 1H), 3.81 (t, *J* = 9.6 Hz, 1H), 2.18 (s, 3H), 2.17 (s, 1H), 2.16 (s, 3H), 2.15 (s, 1H), 2.13 (s, 3H), 2.12 (s, 1H), 2.10 (s, 1H), 2.06 (s, 3H), 2.06 (s, 3H), 2.05 (s, 3H), 2.04 (s, 1H), 2.03 (s, 1H), 2.01 (s, 3H), 2.00 (S, 1H), 1.97 (d, *J* = 1.7 Hz, 3H), 1.96 (s, 1H); $^{13}$C NMR (100 MHz, cdcl$_3$) δ 170.3, 170.2, 170.1, 170.0, 169.9, 169.6, 169.1, 168.9, 101.1, 88.9, 75.7, 70.9, 70.6, 69.5, 69.3, 69.0, 66.5, 61.4, 60.7, 20.9, 20.8, 20.6, 20.5; HRMS (ESI, M + Na$^+$) calcd for C$_{28}$H$_{38}$O$_{19}$Na 701.1905, found 701.1902.

*4-Methylphenyl 2,3-di-O-acetyl-6-O-benzyl-1-thio-β-D-glucopyranoside* (**4a**). To a solution of compound **3a** (100 mg, 0.22 mmol) and acetonitrile (1 mL), were added triethylsilane (352 μL, 2.2 mmol) and ionic liquid **Ia** (37 mg, 0.11 mmol) in the sealed tube. After stirring for one hour at 25 °C, the reaction mixture was added 1 M tetra-*n*-butylammonium fluoride (TBAF, 2.4 mL, 2.4 mmol) and acetic acid (137 μL, 2.4 mmol) and stirred for one hour. The mixture was diluted with water (30 mL) and extracted with ethyl acetate (3 × 30 mL). The combined organic layers were dried over anhydrous MgSO$_4$, filtered, and concentrated. The residue was purified by chromatography to afford desired product **4a** (91 mg, 90%) as yellow oil. $R_f$ 0.25 (EtOAc/Hex = 1/2); $[\alpha]^{26}_D$ −37.4 (*c* 0.8, DCM); IR (NaCl) ν 3478, 3030, 2920, 1752, 1494, 1373 cm$^{-1}$; $^1$H NMR (400 MHz, CDCl$_3$) δ 7.73–7.28 (m, 6H), 7.06 (d, *J* = 8.4 Hz, 2H), 5.04 (t, *J* = 9.2 Hz, 1H), 4.88 (t, *J* = 9.6 Hz, 1H), 4.61 (d, *J* = 10.0 Hz, 1H), 4.59–4.50 (m, 2H), 3.83–3.74 (m, 2H), 3.70 (td, *J* = 9.6, 2.4 Hz, 1H), 3.55–3.51 (m, 1H), 2.92 (s, 1H), 2.30 (s, 3H), 2.07 (s, 3H), 2.05 (s, 3H); $^{13}$C NMR (100 MHz, CDCl$_3$) δ 171.2, 169.5, 138.4, 137.6, 133.4, 129.6, 128.4, 128.0, 127.8, 127.6, 85.8, 78.4, 76.7, 73.7, 70.0, 69.9, 69.87, 21.1, 20.8; HRMS (ESI, M + Na$^+$) calcd for C$_{24}$H$_{28}$O$_7$SNa 483.1453, found 483.1446.

*4-Methylphenyl 2,3,6-tri-O-benzyl-1-thio-β-D-glucopy-ranoside* (**4b**). To a solution of compound **3b** (100 mg, 0.18 mmol) and acetonitrile (1 mL), were added triethylsilane (287 μL, 1.8 mmol) and ionic liquid **Ia** (31 mg, 0.09 mmol) in the sealed tube. After stirring for 24 hours at 25 °C, the reaction mixture was added 1 M tetra-*n*-butylammonium fluoride (TBAF, 2 mL, 2 mmol) for one hour. The mixture was diluted with water (30 mL) and extracted with ethyl acetate (3 × 30 mL). The combined organic layers were dried over anhydrous MgSO$_4$, filtered, and concentrated. The residue was purified by chromatography to afford desired product **4b** (85 mg, 85%) as yellow oil. $R_f$ 0.25 (EtOAc /Hex = 1/2); mp 62–64 °C; $[\alpha]^{26}_D$ −7.4 (*c* 0.8, DCM); IR (NaCl) ν 3478, 3030, 2920, 1752, 1494, 1373 cm$^{-1}$; $^1$H NMR (400 MHz, CDCl$_3$) δ 7.48 (d, *J* = 8.0 Hz, 2H), 7.43 (d, *J* = 7.6 Hz, 2H), 7.39–7.28 (m, 13H), 7.06 (d, *J* = 8.0 Hz, 2H), 4.93 (dd, *J* = 10.8, 6.0 Hz, 2H), 4.77 (dd, *J* = 17.0, 10.6 Hz, 2H), 4.64 (d, *J* = 9.2 Hz, 1H), 4.62–4.52 (m, 2H), 3.83–3.73 (m, 2H), 3.65 (t, *J* = 9.2 Hz, 1H), 3.54 (t, *J* = 8.8 Hz, 1H), 3.50–3.43 (m, 2H), 2.62 (s, 1H), 2.32 (s, 3H); $^{13}$C NMR (100 MHz, CDCl$_3$) δ 138.4, 138.2, 138.1, 138.0, 137.9, 137.7, 132.6, 129.7, 129.6, 128.6, 128.4, 128.37, 128.2, 127.9, 127.7, 87.9, 86.1, 80.4, 78.0, 75.5, 75.3, 73.6, 71.6, 70.3, 21.1; HRMS (ESI, M + Na$^+$) calcd for C$_{34}$H$_{36}$O$_5$SNa 579.2181, found 579.2184.

*Methyl 2,3-di-O-acetyl-6-O-benzyl-α-D-glucopyranoside* (**4c**). To a solution of compound **3c** (100 mg, 0.27 mmol) and acetonitrile (1 mL), were added triethylsilane (431 μL, 2.7 mmol) and ionic liquid **Ia** (47 mg, 0.14 mmol) in the sealed tube. After stirring for one hour at 25 °C, the reaction mixture was added 1 M tetra-*n*-butylammonium fluoride (TBAF, 3 mL, 3 mmol) and acetic acid (171 μL, 3 mmol) for one hour. The mixture was diluted with water (30 mL) and extracted with ethyl acetate (3 × 30 mL). The combined organic layers were dried over anhydrous MgSO$_4$, filtered, and concentrated. The residue was purified by chromatography to afford desired product **4c** (88 mg, 88%) as yellow liquid. $R_f$ 0.35 (EtOAc/Hex = 1/1); $[\alpha]^{26}_D$ +109.5 (*c* 1.0, DCM); IR (NaCl) ν 3474, 2919, 2871, 1747, 1452, 1371 cm$^{-1}$;

[1]H NMR (400 MHz, CDCl$_3$) δ 7.35–7.25 (m, 5H), 5.28 (t, *J* = 9.4 Hz, 1H), 4.88 (d, *J* = 3.6 Hz, 1H), 4.83 (dd, *J* = 10.6, 3.6 Hz, 1H), 4.57 (q, *J* = 12.0 Hz, 2H), 3.81–3.67 (m, 4H), 3.37 (s, 3H), 2.06 (s, 3H), 2.05 (s, 3H); [13]C NMR (100 MHz, CDCl$_3$) δ 176.8, 171.3, 170.2, 137.7, 128.3, 127.6, 127.5, 96.6, 73.5, 72.9, 70.7, 70.1, 70.0, 69.2, 55.1, 20.8, 20.6; HRMS (ESI, M + Na$^+$) calcd for C$_{18}$H$_{24}$O$_8$Na 391.1369, found 391.1362.

*4-Methylphenyl 2,3-di-O-acetyl-6-O-benzyl-1-thio-β-ᴅ-galacopyrano-side* (**4d**). To a solution of compound **3d** (100 mg, 0.22 mmol) and acetonitrile (1 mL), were added triethylsilane (352 µL, 2.2 mmol) and ionic liquid **Ia** (37 mg, 0.11 mmol) in the sealed tube. After stirring for eight hours at 25 °C, the reaction mixture was added 1 M tetra-*n*-butylammonium fluoride (TBAF, 2.4 mL, 2.4 mmol) and acetic acid (137 µL, 2.4 mmol) for one hour. The mixture was diluted with water (30 mL) and extracted with ethyl acetate (3 × 30 mL). The combined organic layers were dried over anhydrous MgSO$_4$, filtered, and concentrated. The residue was purified by chromatography to afford desired product **4d** (73 mg, 72%) as white solid. *R*$_f$ 0.50 (EtOAc/Hex = 1/1); mp 114–115 °C; [α]$^{26}_D$ +1.8 (*c* 0.8, DCM); IR (NaCl) ν 3478, 3030, 2922, 1750, 1494, 1369 cm$^{-1}$; [1]H NMR (400 MHz, CDCl$_3$) δ 7.40 (d, *J* = 8.0 Hz, 2H), 7.37–7.26 (m, 5H), 7.05 (d, *J* = 8.0 Hz, 2H), 5.26 (t, *J* = 10.0 Hz, 1H), 4.94 (dd, *J* = 9.8, 3.0 Hz, 1H), 4.62 (d, *J* = 10.0 Hz, 1H), 4.59–4.48 (m, 2H), 4.15 (d, *J* = 2.4 Hz, 1H), 3.81–3.74 (m, 2H), 3.73–3.67 (m, 1H), 2.77 (s, 1H), 2.29 (s, 3H), 2.07 (s, 3H), 2.05 (s, 3H); [13]C NMR (100 MHz, CDCl$_3$) δ 170.2, 169.5, 164.8, 155.2, 138.3, 137.5, 133.3, 129.6, 128.5, 128.3, 127.9, 127.8, 86.5, 76.7, 74.5, 73.8, 69.5, 68.3, 67.6, 29.7, 21.2, 20.9; HRMS (ESI, M + Na$^+$) calcd for C$_{24}$H$_{28}$O$_7$SNa 483.1453, found 483.1458.

*4-Methylphenyl 3-O-Acetyl-6-O-benzyl-2-deoxyl-2-pht-halimido-1-thio-β-ᴅ-gluco-pyranoside* (**4e**). To a solution of compound **3e** (100 mg, 0.18 mmol) and acetonitrile (1 mL), were added triethylsilane (287 µL, 1.8 mmol) and ionic liquid **Ia** (31 mg, 0.09 mmol) within the sealed tube. Once stirred for one hour at 25 °C, the reaction mixture was added to 1 M tetra-*n*-butylammonium fluoride (TBAF, 2 mL, 2 mmol) and acetic acid (114 µL, 2 mmol) for one hour. The mixture was diluted with water (30 mL) and extracted with ethyl acetate (3 × 30 mL). The combined organic layers were dried over anhydrous MgSO$_4$, filtered, and concentrated. The residue was refined by chromatography to afford desired product **4e** (81 mg, 82%) as white solid. *R*$_f$ 0.38 (EtOAc /Hex = 1/1); mp 54–56 °C; [α]$^{23}_D$ +17.0 (*c* 1.0, DCM); IR (NaCl) ν 3478, 3030, 2922, 2855, 1750, 1494, 1430, 1369 cm$^{-1}$; [1]H NMR (400 MHz, CDCl$_3$) δ 7.83 (t, *J* = 7.6 Hz, 2H), 7.72–7.66 (m, 2H), 7.41–7.27 (m, 7H), 7.00 (d, *J* = 8.0 Hz, 2H), 5.72–5.64 (m, 2H), 4.65–4.52 (m, 2H), 4.26 (t, *J* = 10.4 Hz, 1H), 3.83–3.76 (m, 4H), 3.05 (s, 1H), 2.26 (s, 3H), 1.88 (s, 3H); [13]C NMR (100 MHz, CDCl$_3$) δ 171.1, 167.8, 167.3, 138.4, 137.7, 134.3, 134.1, 133.5, 131.6, 131.2, 129.6, 128.4, 127.8, 127.7, 127.5, 123.6, 123.5, 83.2, 78.3, 74.3, 73.7, 71.0, 70.2, 53.6, 52.6, 21.1, 20.7, 20.5, 13.9; HRMS (ESI, M + Na$^+$) calcd for C$_{30}$H$_{29}$NO$_7$SNa 570.1562, found 570.1561.

## 4. Conclusions

In summary, the Brønsted–Lowry acidic ionic liquid is a particularly economical catalyst for per-*O*-acetylation and ring-opening reactions of sugars. This approached lacks the completely catalytic amount of the least costly accessible ionic liquid that is water-stable. However, the per-*O*-acetylation reactions were conducted under solvent-free conditions, employing a ratio load of acetic anhydride that grants an efficient per-*O*-acetylation reaction of hexoses, and also allows reductive ring opening of benzylidene acetals reactions to take place very evenly. We are still working on the reusability of the reductive ring opening of benzylidene acetals. The reaction conditions are mild, convenient, and nonhazardous for the above reactions.

**Supplementary Materials:** The following are available online at http://www.mdpi.com/2073-4344/10/6/642/s1, Figure S1. Acidity of different ionic liquids; Table S1. For Calculation and comparison of different ionic liquids **Ia–If**.

**Author Contributions:** Conceptualization, S.-Y.L. and W.-Y.H.; analyzed the data, H.-R.W.; methodology, M.T. and Y.-P.W.; software, M.T. and Y.-P.W.; validation, M.T., Y.-P.W., Y.-J.L. and S.-L.D.; writing—original draft preparation, M.T.; writing—review and editing, S.-Y.L.; supervision, S.-Y.L. All authors have read and agreed to the published version of the manuscript.

**Funding:** This research was funded by National Chung Hsing University, and Chia Nan University of Pharmacy and Science.

**Acknowledgments:** The authors thank the Ministry of Science and Technology in Taiwan (MOST 106-2113-M-005-007 and 107-2113-M-005-021 for S.-Y. Luo and MOST 106-2221-E-041-004 for W.-Y. Ho), National Chung Hsing University, and Chia Nan University of Pharmacy and Science for financial support.

**Conflicts of Interest:** The authors declare no conflict of interest. The funders had no role in the design of the study; in the collection, analyses, or interpretation of data; in the writing of the manuscript, or in the decision to publish the results.

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
