# Peer review of "Ionic Liquid Catalyzed Per-O-Acetylation and Benzylidene Ring-Opening Reaction"

_catalysts, doi:10.3390/catal10060642_

Round 1

Reviewer 1 Report

In this manuscript, the authors report ionic liquid catalyzed per-o-acetylation and benzylidenes ring-opening reaction. As mentioned by the authors, catalyst seems robust and maintained activity after repeated use in subsequent reactions.

Specific comments:

  1. Table 1; while screening ionic liquids, with Ia and Ib (entries 1 and 2), the temperature used was 80 oC however with Ic-f (entries 3-6), the temperature was 25 oC. The authors should explain why they used two different temperatures in screening ionic liquids and on what bases they selected the best condition to show substrates scope. In addition, in Table 2, synthesis of compounds 2a-f, the temperature listed was 25 oC, however the synthetic producer of 2a-f the temperature used was 30 oC. The question is why the authors ran reactions at 30 oC instead of 25 oC.
  2. Table 3; while screening ionic liquids, with Ia-c (entries 1, 2, and 3), the solvent used was DCM however with Id-f (entries 4-6), the solvent used was ACN. The authors should explain why they switched solvent in the middle of the ionic liquids screening. The authors should also explain why ACN gave good results compare to DCM and did they try other solvents?
  3. Table 4, in the synthesis of compound 4b-e, the time given in the table and in the synthetic procedures did not match. Please correct the numbering for products in table 4. 

Author Response

Dear sir,

  Thank you

Reviewer 2 Report

The paper of Shun-Yuan Luo et al on per-o-acetylation of carbohydrate and ring-opening of benzylidene acetals using aryl imidazolium ionic liquid cannot, in the present form, be accepted as a paper in catalysts.
No data is given on the ionic liquids 1a to 1f. I couldn’t found references in the text, nor in their previous publication in Molecules (DOI:10.3390/molecules25020352) dealing with the preparation, and characterization of these ionic liquids:
- A general procedure for the synthesis of these unique ionic liquids is clearly needed.
- A full characterization is also necessary (viscosity at 25°C, moisture content), UV spectra, …
The price could be also very interesting.

No comparison was made with classical commercial ionic liquids, like dicyanamide based ionic liquid. Is it worth the time it takes to synthesize (or the price)?
Furthermore, the ratio substrate/ionic liquid has not been studied.
The same is true for the reductive ring opening of benzylidene acetals. Can the yield be improved by changing the ratio substrate/ionic liquid and/or the temperature?

The two plausible mechanisms are classical mechanism and it doesn't help the reader to understand the catalytic effect of aryl imidazolium ionic liquid.

No information on the recycling procedure could be found in the Materials and Methods section. What is the recovery yield of ionic liquid? Working with 37 mg of ionic liquid could be a problem. From figure 2, one could easily see that the yield is decreasing. Could this be explained by degradation or by metathesis? Characterization of the recycled ionic liquid (UV, IRFT, and NMR) could be added in the supporting information.
The results present by Shun-Yuan Luo et al on per-o-acetylation of carbohydrate and ring-opening of benzylidene acetals using aryl imidazolium ionic liquid are interesting but in the present form, could not be accepted in Catalysts.

Author Response

Dear sir,

Thank you

Round 2

Reviewer 1 Report

Please correct the authors information and their affiliations in the supporting information.

Author Response

Dear sir,

Thank you for your important comments. I correct the authors information and their affiliations in the supporting information.

Reviewer 2 Report

Many thanks to the authors to give me information about the preparation and characterization of these ionic liquids. But it would be better to give this to the readers of Catalysts (in the supporting information). These ILs are original or have already been described? The authors should discuss this point in the text.
I assume that with an 80% yield for the first step, the product is purified? How? The authors should provide NMR data and mass spectra (again in the supp. info).
The counter-ion should be added, I-?
How the metathesis is performed and control? The authors should provide NMR data and mass spec. and iodometric titration (again in the supp. info).
I agree with the authors that these ILs catalysts have not been previously used in this transformation. But did the authors test another commercially available ILs for this reaction?

I have noticed that different ratio has been used for the acetylation reaction and the benzylidene reductive ring-opening reaction. But why these ratios?

Author Response

Dear sir,

Thank you

Round 3

Reviewer 2 Report

As a reviewer, I asked for information on the synthesis of the new Ils. The authors send me the information, which is appreciated, but I think that all the readers of Catalysts interested in per-o-acetylation of carbohydrates will be also interested in the synthesis and characterization of these new ILs. Thus, the synthesis of the ILs should be included in the section “3. Materials and Methods”.

In the synthesis of the ILs send to the reviewer only, there is a mistake. The aryl imidazoles 3a to 3d are not protonated. Then, by reaction with a strong acid (TFA, …), protic ionic liquids (PILs) are obtained. References to PILs could also interesting (Chem. Rev. 2008, 108, 1, 206–237; DOI: 10.1021/cr068040u and Phys. Chem. Chem. Phys., 2015,17, 2357-2365; DOI : 10.1039/C4CP04241G)
There is no characterization of the new PILs. NMR? Presence of Cu traces? A dye was used to characterize the acidity. What is the dye? All these data should be added in the section “3. Materials and Methods”.

When working with PILs and in the presence of a large amount of acetate as the by-product of the acetylation with Ac2O, how did the authors checked the absence of metathesis? 19F NMR of the products and recovered PILS 1a could be interesting, especially for figure 2 (recycling of the PILs).

Author Response

Dear sir,

Thank you
